# Implementing substance use services into acute care settings for pregnant and birthing people: A systematic scoping review of implementation and quality improvement strategies

Carla King[1]*, Adetayo Fawole[1], Gregory Laynor[2], Jennifer McNeely[1], Mishka Terplan[3], Matthew Lee[1], Sugy Choi[1]

**1** Department of Population Health, NYU Grossman School of Medicine, New York, New York, United States of America, **2** Health Sciences Library, NYU Grossman School of Medicine, New York, New York, United States of America, **3** Friends Research Institute, Baltimore, Maryland, United States of America

* carla.king@nyulangone.org

## Abstract

Pregnant and birthing people with untreated substance use disorders (SUDs) face multiple risks of mortality and morbidity. Acute care settings (i.e., hospital inpatient, labor/delivery and emergency departments) are one opportunity to provide substance use services, but have had limited implementation. This scoping review synthesized studies that used an implementation science or quality improvement (QI) strategy to implement substance use services into acute care settings for pregnant or birthing people. Our aim was to 1) characterize the implemented strategies; 2) assess the inclusion of racial equity in study design and implementation; 3) summarize measures and outcomes used to evaluate implementation; and 4) identify reported barriers and facilitators to implementation. We searched MEDLINE (PubMed), CINAHL Complete (EBSCO), Scopus (Elsevier), and APA PsycINFO (Ovid) for published studies using keywords and structured vocabulary, and supplemented database searches with a grey literature search of conference proceedings. Two authors independently screened then extracted studies that met eligibility criteria. After removing 661 duplicates, we screened 1101 studies by title and abstract and excluded 1037. Thirty-six were excluded after full text review yielding 28 studies for extraction. Studies were observational (n = 20, 71%), QI (n = 7, 25%), and experimental (n = 1, 4%). Twenty (71%) focused on SUDs broadly; 8 (29%) targeted OUD. Five strategy types were identified: 1) education and learning collaboratives (n = 11, 39%); 2) clinical workflows and pathways (n = 7, 25%); 3) brief interventions (n = 2, 7%); 4) peer support (n = 4, 14%); and 5) structural changes (n = 4, 14%). Five studies (18%) considered racial and ethnic equity in design or implementation. Overall, studies highlight promising strategies to implement substance use services for pregnant and birthing people in acute care settings. However, many strategies were not rigorously evaluated and few considered racial and ethnic equity in design or implementation. Future research

**Data availability statement:** All data analyzed in this review are from previously published studies, which are cited throughout the manuscript. No new data were generated.

**Funding:** CK was funded in part by the Canadian Institutes of Health Research (CIHR) Doctoral Foreign Study Award (Institute of Neurosciences, Mental Health and Addiction, DFD-187710). The funders did not play any role in study design, data collection and analysis, decision to publish, or preparation of the manuscript.

**Competing interests:** The authors have declared that no competing interests exist.

should focus on more rigorous evaluations of implementation strategies, measure downstream outcomes such as adoption and sustained use of substance use services, and apply a racial equity lens more explicitly.

## Introduction

Pregnant and birthing people with untreated substance use disorders (SUDs) face multiple risks of mortality and morbidity, including overdose, severe maternal morbidity, and other adverse pregnancy outcomes [1–4]. Mental health conditions, including overdose or poisonings, are a leading cause of pregnancy-related deaths in the United States (US), and SUDs are estimated to be a contributing circumstance in 25% of cases [1,2]. Rates of severe maternal morbidity and adverse pregnancy outcomes are significantly higher among individuals with SUDs compared to those without SUDs [3–5], and are also highest among Black and Hispanic individuals–both independently and within populations with SUDs–compared to White individuals [4,6]. Severe maternal morbidity and adverse pregnancy outcomes associated with SUDs include cardiac arrest, sepsis, hemorrhagic shock, antepartum hemorrhage, hypertensive disorders, preterm birth, and fetal growth restriction [3,4].

There are existing clinical guidelines [7–11] that provide healthcare settings and providers with recommendations for substance use treatment among pregnant and birthing people, inclusive of alcohol, opioids, stimulants, cannabis, hallucinogens, inhalants, sedatives, and hypnotics. Still, only an estimated 6–13% of reproductive age individuals with SUDs receive SUD treatment [12–14]. Healthcare providers, including obstetricians-gynecologists (OB/GYN) and other prenatal care providers, often report feeling inadequately trained in perinatal SUD treatment and available resources [15,16] and few OB/GYN are trained to provide buprenorphine, an important medication to treat opioid use disorder (OUD) [17].

In addition to provider-level barriers, structural racism worsens gaps in access to and quality of substance use services [18–21]. Minoritized racialized pregnant people are more likely to receive an OUD diagnosis later in pregnancy compared to White pregnant people [22]. and Black and Hispanic people, pregnant and not pregnant, receive SUD treatment and medications for OUD at a lower rate than White people [21,23–26]. Pregnant people with SUDs who are racialized report more intense scrutiny and mistrust from healthcare providers compared to their White counterparts and require greater self-advocacy to access treatment and maintain custody of their children [27].

Acute care settings, including hospital inpatient, labor/delivery, and emergency departments (EDs), are one key opportunity to improve equitable access to substance use services (i.e., medications, referrals to treatment, and overdose prevention) for pregnant and birthing people with SUDs [28,29]. Embedding substance use services into acute care settings could improve the quality of hospital-based care, build patient-provider trust, and improve SUD treatment linkage [30–32]. For perinatal populations, improving access to substance use services in acute care settings also has the potential to lessen racially inequitable treatment gaps [7,15,33].

Implementation and quality improvement (QI) strategies are commonly used in studies that aim to promote implementation and adoption of clinical recommendations by health systems and providers through education, updated clinical workflows and tools, and broader structural changes [34,35]. For the perinatal population in acute care settings, there are no existing reviews summarizing the implementation or QI strategies that have been used to promote implementation or adoption of substance use services by healthcare providers or systems, including how they considered racial equity in the design, or how they measured success. While prior scoping reviews have focused on acute care interventions and transitions of care from the hospital, these studies described strategies for a general population of people with SUDs and did not consider the unique context of pregnancy [36,37]. Other systematic reviews examined interventions that targeted neonatal acute care [38,39] or outpatient clinical settings only [40]. A comprehensive review of the literature is needed to characterize studies that have used a strategy aiming to implement substance use services for pregnant and birthing people into acute care settings, while also considering racial equity.

To address this gap, we used scoping methods to systematically review studies that used an implementation or QI strategy to integrate substance use services for pregnant or birthing people into an acute care setting. In this study, acute care settings include hospital inpatient, labor/delivery, and EDs only. Scoping methods were chosen to review a body of research that is in its early stage of development, and to identify gaps in measurement and outcomes that would aid future research and implementation [41]. We aimed to 1) characterize and describe the implemented strategies; 2) assess the inclusion of racial equity in study design and implementation; 3) summarize measures and outcomes used to evaluate implementation; and 4) identify reported barriers and facilitators to implementation, sustainability, and/or scale-up.

## Methods

This study is reported in accordance with Preferred Reporting Items for Systematic Reviews and Meta-Analyses extension for scoping reviews (PRISMA-ScR) [41]. (see S1 File). The full protocol (registered at OSF, registration number: BC4VZ) was developed using JBI methodology [42] and was previously published [43].

### Eligibility criteria

**Inclusion.** We included US-based studies published between 2016–2023 that described or evaluated a strategy that aimed to implement substance use services into an acute care setting for pregnant and birthing individuals. Substance use services are defined as medications, referrals to treatments, and overdose prevention. The period from 2016 to 2023 captures years of more recently published recommendations and guidelines for SUD treatment practices [7–11], and the launch of the National Network of Perinatal Quality Collaboratives (NNPQC) [44]. We defined SUDs as those involving alcohol, opioids, stimulants, cannabis, hallucinogens, inhalants, sedatives, or hypnotics/anxiolytics. Acute care settings included EDs, labor/delivery, and hospital inpatient only. "Strategies" were broadly defined by implementation science and QI as "methods to enhance the adoption, implementation, sustainment, and scale-up of an innovation" [45] and "interventions or tools" that "systematically improve care", respectively [46].

**Exclusion.** We excluded studies where the strategy did not focus specifically on substance use services for treatment or treatment engagement in acute care settings, including studies that focused primarily on screening for suspected substance use, outpatient care, pain protocols, or neonatal care. Studies that only examined the pre-implementation phase were also excluded. We did not include review articles, commentaries, or dissertations; however, we searched for original research in the references of relevant articles.

### Search strategy

We searched MEDLINE (PubMed), CINAHL Complete (EBSCO), Scopus (Elsevier), and APA PsycINFO (Ovid) for published studies. We developed a core search strategy using keywords and structured vocabulary in MEDLINE (PubMed), then translated the core search for the other databases using the Polyglot Translator [47]. (see protocol paper for more details [43]). We used EndNote 20.6 [48] to manage search results. The final search was conducted in January 2024.

We supplemented our database searches with a targeted grey literature search of conference proceedings to capture upcoming publications. One reviewer (CK) conducted a manual keyword search of conference proceedings published from 2016 to 2023 from two annual US conferences – Association for Multidisciplinary Education and Research in Substance use and Addiction (AMERSA) and Annual Conference on the Science and Dissemination and Implementation in Health. While our protocol indicated we would search conference proceedings from a third conference – Society for Maternal-Fetal Medicine – these proceedings were indexed on Scopus, and so were included as part of the database search. We used a keyword search to find conference abstracts that included terms "pregnant", "pregnancy", "postpartum", "birth", "substance", and "disorder". In February 2024, we consulted with subject matter experts at NNPQC to locate any additional studies or published project reports that met inclusion criteria.

## Screening

All published studies were collated and uploaded to Covidence [49], and duplicates removed. Titles and abstracts, then full texts, were independently screened by two reviewers (CK, AF) using the eligibility criteria. Reviewers met regularly to ensure interrater reliability and resolve disagreements.

Conference abstracts were screened using the same eligibility criteria. One reviewer (CK) initially screened abstracts using the eligibility criteria, then imported those that met criteria to Covidence [49]. A second reviewer (AF) screened the imported conference abstracts. Disagreements were resolved through discussion.

## Data extraction

The study protocol [43] provides further details on the extraction process. Briefly, two reviewers (CK and AF) independently extracted study data using an extraction tool developed as part of the study protocol; minor changes to the tool were made after the first few extractions (see S2 File). We used the same extraction template for published studies and grey literature. Extraction disagreements were resolved through discussion. The extraction tool included a description of the study characteristics, strategy, and outcomes.

We adopted Leeman et al.'s classification framework to broadly classify each study's strategy as a: *dissemination strategy* "target[ing]... awareness, knowledge, attitudes, and intention to adopt an [evidence-based intervention (EBI)]"; *integration strategy*, "...applied to integrate a specific EBI into practice"; *implementation process strategy* that "pertain[ed] to processes or activities that implementation or quality improvement teams perform to plan, select, and integrate an EBI into practice"; *capacity-building strategy* that is "delivered by support systems and target individuals' general capacity...to execute implementation process strategies...across multiple settings"; or *scale-up strategy* "...enacted by support system actors with the goal of getting multiple settings to implement a specific EBI" [50].

Our outcomes of interest were 1) implementation; and 2) the extent to which a racial and ethnic equity lens was incorporated into the study design and implementation. We extracted reported implementation outcomes, as defined by Proctor et al., that related to implementation of substance use services for pregnant and birthing people, including acceptability, feasibility, adoption, and sustainability (see S2 File) [51]. We also noted the level at which the outcome was measured, when applicable (i.e., patient-, provider-, hospital-, or community-level). To assess how studies acknowledged and addressed racial and ethnic equity in their design and implementation, we extracted explicit or implicit mentions of racial equity using Shelton et al.'s recommendations for addressing racism in implementation science as a guide [52]. Reviewers extracted any paragraphs or sentences in the studies where the authors mentioned racial and ethnic equity using terms or phrases such as "equity", "structural racism", "structural competency" or acknowledged existing racial disparities. Reviewers coded these phrases as *explicit* when authors used the specific equity terms above, or *implicit* when the authors only acknowledged racial disparities. Reviewers resolved any discrepancies via discussion.

 

## Data synthesis

We used Callaghan-Koru's updated Model for Improvement, Leeman's strategy classification framework, and the Expert Recommendations for Implementing Change (ERIC) framework to help organize the results [34,50,53]. For each study, we considered the proposed clinical practice changes, and the strategies used to help providers make those changes. Studies that were implemented across multiple settings commonly used both scale-up and capacity building strategies, so we grouped these in the results. We used the updated Consolidated Framework for Implementation Research (CFIR) to categorize the study's barriers and facilitators to implementation, sustainability, and scale-up into one of five domains: 1) the innovation (i.e., the "thing" being implemented), 2) among individuals (i.e., healthcare providers directly or indirectly delivering the innovation), 3) implementation process (i.e., "the activities and strategies used to implement the innovation"), 4) inner setting (i.e., the hospital), and 5) outer setting (i.e., the community) [54].

## Results

### Search results

The database search identified 1757 studies, with an additional six abstracts from conference proceedings added manually (Fig 1). One conference abstract was merged with its full-text study, yielding 1763 references as 1762 studies. After removing 661 duplicates, 1101 studies were screened for eligibility by title and abstract; 1037 were excluded because they did not meet the inclusion criteria (i.e., ineligible study types with no intervention or strategy, ineligible settings, or ineligible interventions). Of the remaining 64 studies, 36 were excluded after full text review: 17 in ineligible settings (i.e., outpatient settings, SUDs treatment programs), seven using ineligible interventions (i.e., neonatal-focused), and 12 with ineligible study designs (i.e., descriptions of strategies that had not been implemented). Studies recommended by NNPQC subject matter experts that met eligibility criteria had already been included, yielding 28 studies in the final analysis.

### Study characteristics

Table 1 summarizes characteristics of the 28 studies included in the review. Studies were observational (n = 20, 71%), QI projects (n = 7, 25%), and experimental (n = 1, 4%). Most were published in 2022 (n = 9, 32%) and 2023 (n = 7, 25%). Seven (25%) used a QI or implementation science framework or model to guide study design and methodology. All major US census regions were represented by the studies, with most strategies implemented in the South (n = 10, 36%) and West (n = 8, 29%) regions (Fig 2). Nine studies (32%) explicitly involved rural hospitals or providers. Many studies (n = 13, 46%) were implemented across multiple hospital settings or among providers broadly, while others specifically targeted labor/delivery/neonatal intensive care units (n = 11, 39%). Twenty studies (71%) focused on people with SUDs broadly, while other studies (n = 8, 29%) targeted people with OUD only.

### Strategies and outcomes

We grouped studies into five types according to the implementation/QI strategy used to integrate substance use services: 1) education and learning collaboratives; (n = 11, 39%); 2) clinical workflows and pathways (n = 7, 25%); 3) brief interventions (n = 2, 7%); 4) peer support (n = 4, 14%); and 5) structural changes (n = 4, 14%). In Table 2, we provide a summary of each study by type (a more detailed table is in S3 File). A brief description of the strategy with examples, relevant outcomes, barriers and facilitators, and attention to racial equity are discussed in more detail below.

**Education and learning collaboratives.** Eleven studies (39%) used seminars, fellowships, or learning collaboratives to improve provider knowledge about SUD treatments and the delivery of compassionate care for pregnant and birthing people with SUDs [55–65]. Four studies (14%) targeted nursing and medical students/trainees [59,62–64]. We classified six of the 11 studies (55%) as dissemination strategies and five (45%) as scale-up and capacity-building strategies.

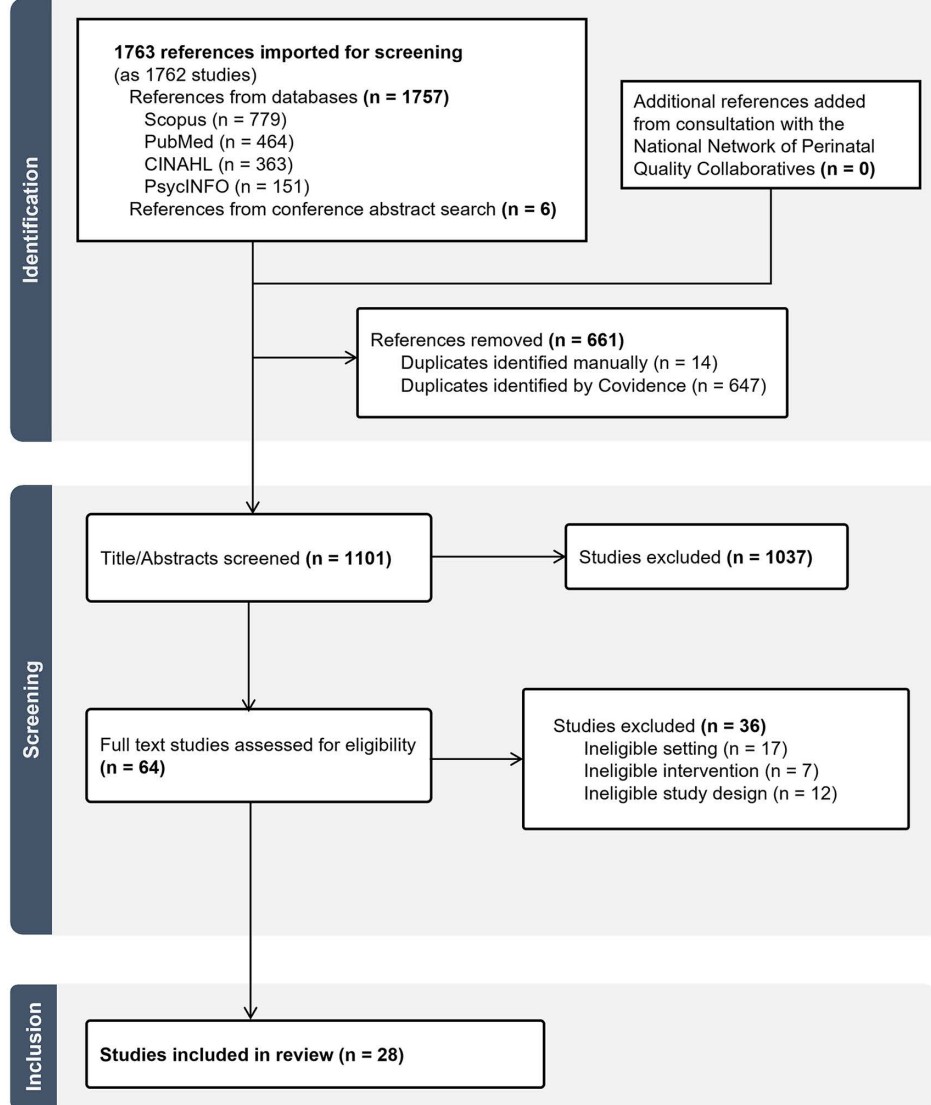

**Fig 1. PRISMA flow diagram.**

Six dissemination studies described brief educational sessions with healthcare providers [55,57–60,64]. These studies delivered a single seminar or presentation; some included an experience-based component (i.e., role playing, simulation) [59,60,64]. For example, Shuman et al. recruited nursing school students to pilot an arts-based intervention that used creative writing and art to improve attitudes towards pregnant people with SUDs [59].

Three studies provided longer-duration education for healthcare providers [56,61,65]. In these studies, providers attended multiple educational sessions on topics including implicit bias, perinatal co-morbidities with SUDs, and use of medications for addiction treatment during pregnancy. One study used the ECHO learning collaborative model to target rural providers who care for perinatal patients with SUDs [61], while another initiated a Maternal Speakers Bureau to disseminate knowledge on OUD during pregnancy across the state [65]. Maternal Speakers Bureau participants completed a 10-week training course on delivery of standardized evidence-informed presentations, and then leveraged their internal network of healthcare providers and hospital staff for dissemination [65].

**Table 1. Characteristics of Included Studies.**

| | N = 28<br>N (%) |
|---|---|
| **Study Design** | |
| Experimental/ randomized trials | 1 (4) |
| Observational | 20 (71) |
| Quality improvement (QI) projects | 7 (25) |
| **QI or implementation science model/ framework used** | 7 (25) |
| **Healthcare setting** | |
| Emergency Department | 1 (4) |
| Inpatient (excluding Labor/ Delivery/ NICU) | 3 (11) |
| Labor/ Delivery/ NICU | 11 (39) |
| Multiple | 13 (46) |
| **Rural hospitals/ providers included** | 9 (32) |
| **SUDs targeted** | |
| Any substances | 20 (71) |
| Opioids only | 8 (29) |
| **Racial equity lens[a]** | 5 (18) |
| Explicit | 4 (14) |
| Implicit | 1 (4) |

NICU, neonatal intensive care unit; SUDs, substance use disorders.

[a]Explicit mentions of racial equity described or named "racial equity", "structural racism", "structural competency" anywhere in the study, whereas implicit mentions of racial equity acknowledged that racialized disparities exist but did not use equity-specific language.

Two studies targeted medical education [62,63]. For example, Laks et al. developed a two-year women's health track within their accredited addiction medicine fellowship [62]. The fellowship provided specialized training in perinatal SUD and SUD treatments through clinical rotations, didactic curriculum, research, quality improvement initiatives, and advocacy work [62].

Most studies reported acceptability-related implementation outcomes by measuring changes in healthcare provider knowledge, stigma, and attitudes towards pregnant and parenting people with SUDs and SUD treatment; some used adapted knowledge and stigma scales to measure change including Attitudes of Health Care Providers, The Opening Mind Scale for Healthcare Workers, and modified Attitudes About Drug Use in Pregnancy Scale [55–59]. Comparing pre-post surveys, there were reported improvements in stigma scores, knowledge scores, and attitude scores after educational interventions among physicians, nurses, and social workers [56–60]. For example, Walsh et al. reported that obstetric providers showed decreased mean stigma scores ($p < .015$) and increased mean compassion ($p < .036$), knowledge ($p < .001$), and attitude scores ($p < .006$). Only one study examined sustainability and found that healthcare providers' perceptions towards pregnant people with SUDs improved directly after an educational intervention (mean adapted attitudes of healthcare providers survey scores: time 1 mean(SD) =38.24(8.93), time 2 mean(SD)=32.71(7.77), $p < 0.05$), however, changes were not sustained after 60 days (time 3 mean(SD)=37.08(8.45), $p < 0.05$) [55].

Other studies reported acceptability and feasibility measures at a single time [61–63,65]. For example, Hostetter et al. reported that their experience-based curriculum was both feasible and acceptable by medical students and educators [63], but did not measure adoption or intended use of the newly acquired knowledge or skills. Laks et al. reported that their fellowship led to new partnerships in the community, collaboration between OB/GYN and addiction medicine, and that participating fellows contributed to local QI and advocacy projects [62]. Moore et al. reported that the ECHO

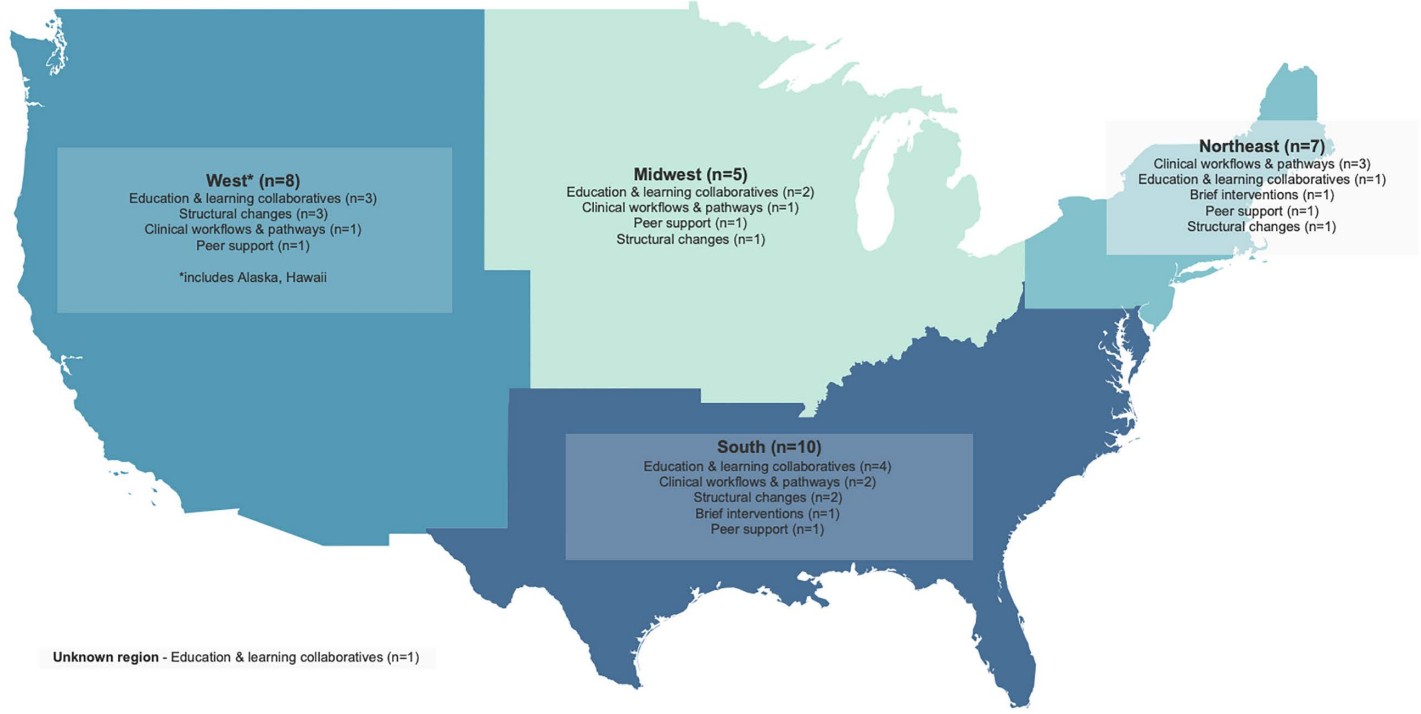

**Fig 2. Map of included studies by US census regions.** Darker colors represent regions with more studies. Strategy types are listed in order of the most commonly used in that region. One study (classified as using a 'structural changes' strategy) had participants from all regions and so was included in the count for each. [Map made with Tableau and Natural Earth. Free vector and raster map data @ naturalearthdata.com].

series was 'reachable' and well attended, and that participants felt better prepared to treat people with perinatal SUDs after attending [61].

**Clinical pathways and workflows.** Seven studies (25%) implemented clinical pathways or workflows to address hospital-level gaps in SUD treatment, referrals, and overdose prevention [66–72]. Most strategies targeted providers in labor/delivery units (n = 5/9, 56%) [66,67,69,72] while others focused on emergency departments (n = 1/9, 11%) [68] or acute care settings generally (n = 2/9, 22%) [70,71].

Two studies specifically used clinical pathways to establish collaborative care between outpatient and inpatient providers [66,69]. For example, as part of a broader strategy to improve perinatal substance use care, one study integrated the hospital's outpatient perinatal clinic and inpatient addiction consult team [66]. In partnership with the clinic, the consult team provided withdrawal management, SUD treatment, and care coordination for pregnant and postpartum patients during and after hospitalization. Another study, located in a rural setting, developed a nurse-led community-based program to improve treatment referrals for pregnant people with OUD [69]. As part of the workflow, in-hospital social workers met with patients after delivery to review, discuss, and implement care plans developed by the program's nurse care coordinator.

Three studies introduced clinical pathways via scale-up and capacity-building strategies [68,70,71]. For example, at the state-level, Kilaru et al. implemented an Opioid Hospital Quality Improvement program where hospitals earned payments for designing and implementing clinical pathways, one of which was for post-discharge OUD referrals and treatment for pregnant patients [71].

Clinical workflows were also introduced to provide naloxone kits and overdose education to postpartum people at hospital discharge [67,72]. Naliboff et al. piloted a workflow for universal distribution of a first aid kit with naloxone for patients

**Table 2. Summary of studies from 2016-2023 that used an implementation/QI strategy to integrate substance use services for pregnant or birthing people in acute care settings.**

| Author, Year | Target population[a] | Strategy description | Reported implementation outcomes[b] | Considered racial equity? |
|---|---|---|---|---|
| **1. Education and learning collaboratives** | | | | |
| Ford et al., 2021 | Hospital providers | 32-month multi-site education collaborative that included education of SUD treatments, community/parent outreach and partnerships with local SUD treatment programs | Increased provider attitude scores from baseline to midpoint and mid-point to final assessment at 32-months | No |
| Hostetter et al., 2022 | Students | 10-month experiential learning curriculum for third-year medical students providing exposure to SUD and SUD treatments during pregnancy | Students rated the program highly feasible and acceptable; resources were useful, and the curriculum was a "valuable learning experience" | No |
| Laks et al., 2023 | Hospital providers | Two-year Maternal Health Addiction fellowship with specialized training in perinatal care and addiction medicine | The program was feasible and valuable, and led to improvements in clinical training/education and collaboration between addiction medicine and OB/GYN | Yes – explicitly |
| Merritt et al., 2022 | Hospital providers | One-time educational session at a conference for healthcare professionals about substance use and caring for pregnant people with SUDs | Provider attitudes improved post-intervention; changes not sustained after 60 days | No |
| Moore et al., 2023 | Hospital providers | 10-session ECHO program (30-minute didactic lecture followed by a 30-minute collaborative case review) for rural perinatal providers | The program was "reachable and well-attended"; Participants reported increased knowledge/awareness of SUD treatments and referral resources | No |
| Rudolf et al, 2018 | Hospital providers | 120-minute educational session on substance use in pregnancy, stigma, medications, and compassionate care | Decreased stigma scores, increased compassion, knowledge, attitudes, and comfort with SUD scores | No |
| Shuman et al., 2022 | Students | Arts-based intervention pilot that used creative writing and art to improve provider attitudes and reduce stigma towards perinatal patients | Improved attitudes toward perinatal substance use; participants were highly satisfied | No |
| Stephenson et al., 2022 | Hospital providers | Development of a Statewide Speakers Bureau to disseminate best practices for OUD in pregnancy to local healthcare providers | 57.5% of providers were very confident that the education will improve their care of pregnant and postpartum people; 71% reported being very likely to apply the information to their practice | Yes – explicitly |
| Tobin et al., 2018 | Hospital providers | Two-hour seminar on evidence-based care with role playing | Increased knowledge of addiction and treatment | No |
| Walsh et al., 2017 | Hospital providers | 90-minute education session on withdrawal, treatment, and compassionate care for SUD | Decreased stigma scores and increased compassion, knowledge (including referrals to treatment) and attitudes scores | No |
| Wands et al., 2022 | Students | 20-minute simulation and debrief with a standardized patient portraying a pregnant person with SUD | Improved motivational interviewing skills and caring behaviors | No |
| **2. Clinical workflows and pathways** | | | | |
| Kilaru et al., 2020 | Hospitals | State-funded hospital incentive program where hospitals earned payments for designing clinical pathways for OUD | 76% of hospitals adopted a pathway for post-discharge referrals and treatment for pregnant people with OUD | No |
| Lilly et al, 2019 | Hospital administrators and providers | Collaborative workflow to screen and refer patients to outpatient SUD clinics from labor/delivery and ED | Program reached high-risk groups (including those in labor/delivery) | No |
| Naliboff et al., 2023 | Hospital providers, post-partum people | Developed a workflow for universal distribution of a first aid kit with naloxone | 82% of providers completed educational modules; 97% of postpartum people received overdose education and 76% accepted naloxone | No |

*(Continued)*

**Table 2.** (Continued)

| Author, Year | Target population[a] | Strategy description | Reported implementation outcomes[b] | Considered racial equity? |
|---|---|---|---|---|
| Paterno et al., 2019 | Hospital providers | Nurse-led community-based pilot program to improve SUD screening and referral pathways at a maternal care practice and community hospital | 89.5% of pregnant people received at least one prenatal referral to a community resource post-intervention, compared with 57.9% pre-intervention | No |
| Snyder et al., 2021 | Hospitals | Large-scale implementation project with technical assistance to improve uptake of low-threshold ED buprenorphine | 92.3% of hospitals initiated buprenorphine for pregnant people | No |
| Stone et al., 2023 | Hospital providers | Electronic health record order set for naloxone distribution to postpartum patients at an increased risk of opioid overdose | Naloxone orders at discharge increased from 12.7% to 57.1% | No |
| Townsel et al., 2023 | Hospital providers | Integrated workflow between ACS and outpatient services for pregnant persons with SUDs | ACS initiated 9 hospitalized pregnant people on buprenorphine | Yes – implicitly |
| **3. Brief Interventions** | | | | |
| Shenai et al., 2019 | Pregnant patients | Brief (20–30 minute) intervention for hospitalized pregnant women with a history of substance use and trauma | Improved patient knowledge of substance use and increased likelihood of pursing treatment | No |
| Stotts et al., 2022 | Hospital providers, mothers in NICU | Motivational interviewing, acceptance and commitment therapy (MIACT) intervention for mothers to facilitate SUD treatment initiation | Increased SUD treatment initiation compared to baseline (RR 1.5) | No |
| **4. Peer support** | | | | |
| Gannon et al., 2022 | Hospital providers, people with OUD interacting with doulas | Doula support intervention for pregnant people with OUD | Pregnant people with OUD perceived doulas as acceptable. Pregnant people in treatment for OUD reported less stigma perceived from healthcare providers when a doula was present | No |
| Kivlighan et al., 2022 | Hospital providers | Trained and integrated volunteer birth companions to provide support during pregnancy/labor/delivery, including perinatal substance use education | No reported outcomes | Yes – explicitly |
| Newell et al., 2022 | Hospital providers | Peer support group in the hospital women's health unit for pregnant people with SUDs | Improved patient comfortability sharing SUD-related information with nursing staff after participation | No |
| Schulman, 2020 | Hospital providers | Peer workforce training to provide support to pregnant people with SUDs in the community and during hospitalization, including recovery support, treatment options, and community resources | 640 certified PRCs serving one hospital and the community; 110 encounters and 50 postpartum people continue to engage with PRCs | No |
| **5. Structural changes** | | | | |
| Kroelinger et al., 2019 | State administrators | Federally led learning community supporting 12 states in their development and implementation of system-level changes to OUD care | 83% of state teams developed plans to improve access to and coordination of OUD services; 75% developed plans to address health care provider awareness and training | No |
| Martin et al., 2023 | Hospital providers | Revised hospital policy on in-hospital substance use recommending providers to "respond to substance use concerns by offering patients adequate pain control, evidence-based addiction treatment, and supportive services instead of punitive responses" | No reported outcomes | Yes – explicitly |

*(Continued)*

**Table 2.** (Continued)

| Author, Year | Target population[a] | Strategy description | Reported implementation outcomes[b] | Consid-ered racial equity? |
|---|---|---|---|---|
| Nichols et al., 2018 | Hospital administrators and providers | State-level knowledge transfer collaborative to develop plans for increasing uptake of evidence-based practices related to opioid use during pregnancy | Produced tailored toolkits and best practice guidelines; provided knowledge exchange between researchers, advocates, and practitioners; reported that local coalitions and work groups implemented activities within their counties after participation | No |
| Sharp et al., 2023 | Hospital administrators and providers | Evaluation of the state mandated implemen-tation of CARA which provides supportive, non-punitive SUD care and care coordina-tion for pregnant people with SUDs and their families | 40% of eligible newborns with Medicaid insurance did not have a plan of safe care; 54% of families said someone discussed a plan of safe care with them; 26% were involved in its development | No |

NICU, neonatal intensive care unit; SUDs, substance use disorders; OUD, opioid use disorder; CARA, Comprehensive Addiction and Recovery Act; ACS, addiction consult service; PRCs, Peer Recovery Coaches.

[a]Hospital providers include physicians, nurses, social workers, care coordinators, addiction recovery specialists, certified nurse-midwives, counselors, doulas.

[b]Reported outcomes include those specific to the strategy for SUD care/treatment for pregnant or birthing people. Other outcomes from the studies are not reported.

hospitalized for labor and delivery [72]. Similarly, Stone et al. introduced an order set in the electronic health record (EHR) to increase provider naloxone orders for postpartum patients at an increased risk for opioid overdose [67]. Implementation of naloxone workflows were accompanied by staff educational sessions, a readiness checklist, staff scripts, posters, and information sheets [67,72].

Most studies of clinical pathways or workflows reported early adoption measures at the patient-, provider- or hospital-level [66–69,71,72]. For example, Townsel et al. reported that collaboration between the addiction consult team and outpatient clinic led to nine pregnant people starting buprenorphine during hospitalization [66]. Naliboff et al. reported that nurses provided overdose education to 97% of postpartum people at hospital discharge, and naloxone kits were accepted by 76% [72]. Finally, Kilaru et al. reported that 76% of the hospitals in the incentive program attested to creating a referral and treatment pathway for pregnant patients with OUD [71].

**Brief interventions.** Two studies tested the feasibility of a brief evidence-informed behavioral intervention in the hospital setting to improve linkage to SUD treatment [73,74]. Shenai et al. used an existing treatment manual to provide brief (20–30 minute) education and resources to hospitalized antepartum people with a history of substance use and trauma [73]. Similarly, Stotts et al. integrated motivational interviewing, acceptance and commitment therapy (MIACT) to facilitate SUD treatment initiation among people who used substances during pregnancy and had a hospitalized newborn [74]. While Shenai et al. worked primarily in the hospital's antepartum unit, Stotts et al. focused specifically on people with a neonate in the NICU [73,74].

Both studies reported that patients valued the sessions [73,74]. Stotts et al. used a randomized controlled design comparing the MIACT intervention to usual care. The MIACT group demonstrated an increase in SUD treatment initia-tion during the treatment phase (RR = 1.5) and a significant decrease in risk of child removals by child protective services (CPS) (i.e., a 30% and 10% reduced risk at 2-month and 6-month follow-up, respectively) [74].

**Peer support.** Four studies described the integration or scale-up of peer support, including doulas and volunteer birth companions [75–78]. Peer support provided social and emotional support to pregnant and parenting people with SUDs and helped facilitate communication between pregnant people and healthcare providers [75–78]. Doulas and volunteer birth companions were available during pregnancy and postpartum, in and out of the hospital, in-person and virtually [75–78]. Volunteer birth companions received on-the-job training through online modules, hospital volunteer training,

and a 16-hour live presentation on implicit bias, trauma-informed care, and other relevant topics delivered in part by a community-based reproductive justice organization [76].

Two of the four studies reported acceptability- and appropriateness-related outcomes [75,77,78]. In Gannon's qualitative study, participants reported that the presence of doulas during labor and delivery lessened their experiences of stigma from healthcare providers and improved their health literacy and self-advocacy [75]. Similarly, participants attending a peer support group felt more accepted by nursing staff after disclosing their participation in the program [77]. As of publication in 2020, the peer support training program in Georgia reported a community-level adoption outcome – there were over 640 peers certified and integrated into a local hospital and the surrounding community [78].

**Structural changes.** Four studies used blueprints or policies to structure change [79–82]. Most strategies involved federal and state-level support (i.e., CDC, State Departments of Health, Medicaid and Commercial insurance partners) across multiple hospital systems.

At the local hospital-level, one study revised a policy on in-hospital substance use [79]. The existing policy was deemed "punitive" and "inequitable", and so an interprofessional workgroup of administrators, nurses, and physicians convened to establish a policy change. The revised hospital policy recommended hospital providers respond to patients, including those who are pregnant and racialized, who were using substances during hospitalization by offering pain and withdrawal support, addiction treatments, and supportive services for the SUD.

Three other studies developed or evaluated state-level action plans or policies [80–82]. For example, one study described the use of a learning community to support 12 states in their development and implementation of system-level changes to OUD care, including changes to access and coordination of treatment services, provider awareness and training, and data monitoring/evaluation [80]. Another study evaluated the state's implementation of the Comprehensive Addiction Recovery Act (CARA) and Plans of Safe Care which are intended to provide supportive, non-punitive SUD treatment referrals and care coordination for pregnant people with SUDs and their families [81]. Implementation was supported by the state's Child, Youth and Families Department and Department of Health, and included training for hospital staff and care coordinators on equitable, non-stigmatized care, and development of Plans of Safe Care that help identify SUD treatment referral needs.

Two studies measured early adoption [80,82]. Kroelinger et al. reported that 10 of 12 participating states developed action plans "to address access to and coordination of quality services for pregnant and postpartum women with OUD", and 9 of 12 states focused on improving healthcare provider awareness and training [80]. Sharp et al. completed an implementation evaluation of New Mexico's CARA and made several recommendations to improve provider training and increase the resources allocated to CARA implementation [81]. Sharp et al. found that just over half of families reported someone talked to them about their Plan of Safe Care and about a quarter were involved in its development [81].

## Barriers and facilitators to implementation, sustainability and scale-up

Multiple studies identified barriers and facilitators to implementation, sustainability, and scale-up. Table 3 provides examples in each CFIR domain.

**Program/policy characteristics.** Barriers and facilitators related to the implemented strategy centered around design and complexity. For example, Shuman et al. reported that an arts-based educational program needed to be digital and asynchronous to be implemented at scale [59]. Hostetter et al. reported that individualized learning and one-on-one mentorship was a strength of their addiction curriculum program, however, this limited the number of spots available for training [63]. Finally, Naliboff et. al reported that universal distribution of naloxone, rather than a targeted intervention for people who use drugs, was key to program acceptance among postpartum people [72].

**Implementation process.** Two studies noted that limited time to implement was a barrier to success [65,70]. For example, Stephenson et al. reported that more time for implementation of their educational intervention was needed to help "creat[e] more cohesive implementation of the project across…healthcare facilities". Similarly, Lilly et al. reported

**Table 3. Reported barriers and facilitators to implementation, sustainability, and scale-up by CFIR domains and constructs.**

| CFIR Domain | Barriers | Facilitators |
|---|---|---|
| Program or policy characteristics | Time consuming [81] Synchronous [59] | Universal resource distribution (rather than targeted) [72] |
| Implementation process | Limited time allocated to hiring and recruitment [65,70] | – |
| Healthcare providers | Baseline provider knowledge [64,81] Persistent bias/stigma [79,82] | – |
| Acute care setting | – | Leadership support [56] Provider collaboration [65] Dedicated program staff [62,70] |
| Community | Patient resources (i.e., transportation and telecommunication) [70] Funding and insurance [66,76,81] Rural community treatment access [68,81] | Community involvement/collaboration [56,72,76] |

CFIR, The updated Consolidated Framework for Implementation Research [54].

that establishing a clinical pathway took "more time than anticipated", particularly around hiring and integrating staff, and building relationships between providers.

**Healthcare providers.** Four studies reported barriers to implementation that involved knowledge and attitudes of hospital staff [64,79,81,82]. In one intervention, less clinical experience among students posed a barrier to educating them on more complex care for pregnant people with SUDs [64]. Persistent biases towards pregnant people with SUDs and "ambivalence towards evidence-based recommendations" was a reported barrier to uptake of evidence among community workgroups [82]. Martin et al. noted that healthcare providers had concerns around personal and patient safety when implementing the new policy for in-hospital substance use, however, education and buy-in from hospital leadership, legal, security, and regulatory helped mitigate this barrier [79]. Finally, Sharp et al. noted that minimal provider and hospital training on ICD-10 diagnostic codes limited implementation of CARA [81]. More education and standardization was recommended to reduce bias among hospital staff [81].

**Acute care settings.** Several studies described leadership support, collaborative teams, and existing relationships in the hospital context as key facilitators [56,62,65,70]. For example, Tennessee's Speakers Bureau, led by an interprofessional collaboration of nurses, physicians, and administrators, trained speakers who were already known within local healthcare facilities; this helped enhance support for and uptake of their educational program [65]. Laks et al. reported that existing relationships between Addiction Medicine and OB/GYN educators helped garner program support and led to the supervision of fellows by those not previously involved in the program [62]. Notably, Lilly et al. found that collaborations need to be "developed, nurtured, and maintained to provide effective services" and reported that teams met regularly to improve and sustain communication [70].

Three studies noted the importance of a dedicated staff member to facilitate implementation and sustainability [62,69,70]. For example, Lilly et al. noted that their program required a dedicated staff position to help coordinate and monitor the program and maintain communication with community-based resources [70]. Similarly, Paterno et al. described the importance of a designated nurse coordinator who helped facilitate trust and continuity between women and providers [69]. Lak et al. reported that their program manager was key to organizing and scheduling rotations across addiction medicine and OB/GYN [62].

**Community.** Barriers reported in the outer setting were related to treatment access in the community–rurality, transportation, and childcare. For example, the California Bridge program reported that hospitals in rural areas had difficulty locating follow-up care for patients post-ED visit [68]. Sharp et al. also described limited resources in rural communities, noting that many communities do not have access to inpatient treatment programs that accept pregnant people or people with newborns [81]. Limited telecommunication (i.e., cell phones out of minutes, out of coverage areas) worsened communication issues between providers and patients, however, contact via social media and text messaging was helpful to maintain communication [70].

Several studies noted barriers around funding and insurance. Townsel et al. noted complications related to consult billing and reported that funding for their collaborative workflow without state Medicaid expansion and institutional support would be "extremely challenging" [66]. The lack of insurance reimbursement was also concerning for volunteer birth companions, particularly as it relates to racial equity [76]. Kivlighan et al. reported that while program implementers made efforts to hire staff with diverse and representative backgrounds, not having funds or reimbursement options for birth companions made it difficult to do so [76]. Sharp et al.'s evaluation of CARA implementation reported that "funding dedicated to support the program's needs, which includes multiple full-time positions to deliver program management, case management, training, and reporting system development" is needed to support policy fidelity [81].

Several studies found that community collaboration was a facilitator for implementation [56,72]. For example, Ford et al. found that the program's partnership with parents with SUDs played a critical role in changing provider attitudes [56]. Naliboff et al. reported community and hospital pushback towards universal naloxone distribution initially; however, education for hospital staff coupled with community outreach reportedly improved perceptions of the program [72]. Kivlighan et al. noted that community partnership helped the program develop a "communal vision", "think beyond existing hospital structures" and "build around shared values for reproductive justice and equity" [76].

### Racial equity

Five studies (18%) broadly considered racial equity in design or implementation [62,65,66,76,79].

One study implicitly considered how racialized groups disparately experience CPS involvement. In the development of their clinical pathway, Townsel et al. acknowledged the importance of discussing the involvement of CPS with parents early to ease anxiety from "perceived or actual racial profiling" [66].

Four studies more explicitly considered racial equity. Most integrated racial equity topics into their curricula or training modules, by including sections on intersectionality, racial disparities, and racial bias [62,65,76]. Kivlighan et al. also rooted their volunteer birth companion program in reproductive and birth justice frameworks, structural racism, intersectionality, and community collaboration [76]. Kivlighan et al. noted the importance of representation among service providers and participation by those with lived experience during design and implementation of the program [76]. As another example, Martin et al. adapted their hospital's policy on in-hospital substance use to in part address institutional racism. During implementation, the workgroup took steps to "discuss [the difference between] safety threats versus fears founded in racism and stigma...in which security might be called" and identified the health care system's role in promoting health equity and anti-racism [79].

While most studies did not directly measure outcomes to evaluate the success of including a racial equity lens, Stephenson et al. reported that several participants expressed an awareness of and interest in racial equity and implicit bias training after their education session [65].

### Discussion

Evidence-informed treatment of SUDs during pregnancy is a public health priority [83–85] and acute care settings are an important place to provide effective, potentially life-saving substance use services [15,28,29]. In this scoping review, we characterized implementation and QI strategies used to aid the implementation and adoption of substance use services by

healthcare systems and providers for pregnant and birthing people in acute care settings. We provide a map of potential strategies that could be considered by local and state health systems including 1) provider education; 2) clinical workflows or pathways; 3) brief interventions; 4) peer supports; and 5) structural changes through policy and implementation blueprints. While these strategies varied in scope and audience, they collectively represent early efforts to address persistent gaps in equitable SUD care delivery within acute care environments. Our findings highlight that many interventions focus on shifting provider attitudes and knowledge; however, few studies evaluated whether such changes led to sustained adoption or improved patient outcomes. Additionally, there was wide variability in how implementation outcomes were defined and measured, limiting the ability to compare across studies or draw conclusions about effectiveness. With the urgent need to address SUDs among pregnant and birthing patients, and limited knowledge of SUDs among perinatal providers, local and state health administrators could consider implementing and evaluating a bundle of strategies that may include robust provider and trainee education, clinical workflows and order sets, as well as policies to shift hospital culture and practice. Partnering with community organizations could improve direct care for pregnant people with SUDs through peer support, provide access to provider education networks, and align implementation with the community's needs.

Many education strategies showed promise for improving knowledge of SUD treatments and referrals, as well as attitudes and perceptions of pregnant and birthing people with SUDs among nurses, physicians, and medical trainees. Most studies measured implementation outcomes related to acceptability and feasibility, and found educational sessions to be well-attended and acceptable to participants. The ECHO model, for example, is a well-established strategy for improving knowledge of best practices among primary care providers [86], and indeed, reported improved knowledge and comfort using SUD treatments among rural perinatal providers [61]. Integration of clinical best practices into medical education through case-based, experiential learning, as described in several studies, has also been shown to effectively improve knowledge, attitudes, and skills among medical students [87]. Most studies, however, did not directly measure adoption or uptake of substance use services, such as the use of medications or referral resources in response to the educational program, and one study reported that changes in attitudes and perceptions were not sustained after the educational intervention ended. Short-term provider-level educational interventions alone may not be enough to improve care for pregnant and birthing people with SUDs who face increased discrimination and mistrust within healthcare [15,27,88]. Implementation and QI projects that couple longer-term provider education with other hospital- and system-level supports may be needed to enact and sustain practice change [89].

More direct strategies, such as clinical workflows, peer support, and brief interventions were reportedly acceptable and feasible for pregnant and birthing people with SUDs, and some demonstrated adoption of substance use services in acute care settings. Changes to clinical workflows, for example, showed improvements in SUD treatment and naloxone delivery [66–72], while brief intervention strategies increased linkage to SUD treatment [73,74]. Peer support, including doulas and volunteer birth companions, demonstrated more supportive hospital experiences for pregnant and birthing people with SUD and connections to community-based treatments [75–78]. However, few studies rigorously measured implementation outcomes, with many often reporting a single measure at one level (e.g., adoption by hospitals). For example, the large-scale implementation project for ED buprenorphine, designed for the general population of hospitalized patients with SUDs, reported adoption of buprenorphine for pregnant and birthing people; however, this was limited to a one-time hospital-level measure [68]. Healthcare systems with active implementation and QI projects for the general population of hospitalized patients with SUD could consider more robust measurement of implementation within perinatal spaces and among perinatal providers. Further, future studies would benefit from a more rigorous implementation design that measures outcomes at multiple levels and time points.

While most studies did not focus on sustainability of practice change, several noted the importance of funding, reimbursement, and investment in dedicated program staff or educators. Sustainable change may require long-term investment in personnel to coordinate and monitor progress, build and maintain collaborations, and ensure adequate provider education and training. Insurance reimbursements for consult models or birth companions would help ensure

future viability of programs. Policymakers could also support sustainability by working with payers to improve reimbursement structures and by continuing financial support for federal and state-funded support networks. For example, the NNPQC and state Perinatal Quality Collaboratives, who were involved in several of the studies included in this review [56,65,70,72], are well-positioned to continue building and monitoring local capacity, rapidly scaling-up programs, and supporting hospital systems with resources and data to improve clinical practice [44,90].

Amidst alarming racial and ethnic inequities in maternal mortality and morbidity among people with and without untreated SUDs [3,6,91,92], minimal attention to racial and ethnic equity and structural racism among the studies in this review is concerning; few studies considered racial and ethnic equity in study design or methodology. Among those that did, most targeted individual-level education on implicit bias, intersectionality, or structural racism, and few included a community-engagement component. No studies included specific measures or outcomes, qualitatively or quantitatively, specific to racialized groups. QI and implementation science have an opportunity to advance equity in healthcare by examining, monitoring, and addressing the effects of structural racism and other structural inequities (e.g., socioeconomic status, gender, sexual orientation) on equitable reach, adoption, acceptability, feasibility, and appropriateness of implementation [52,93,94]. From this review, Kivlighan et al. provides one example of practices to consider beyond individual-level education–this study utilized a reproductive justice framework to incorporate a racial equity lens, collaborated with the community before and throughout implementation, and aimed for representation among their workforce [76]. Identifying and measuring existing disparities, integrating structural racism into implementation science and QI frameworks and models, and engaging clinical and community members, particularly those who experience structural racism, are necessary steps towards ensuring equitable implementation [52,93,94]. Locally, hospital systems could consider the addition of hospital-based SUD treatment metrics, disaggregated by a patient's race and pregnancy status, as part of their quality improvement initiatives [93].

Finally, few studies used models or frameworks to guide implementation. Strategies and implementation outcomes were often poorly defined and heterogeneous, making it difficult to compare across studies or to determine mechanisms that made a strategy successful within a specific context. Using existing implementation frameworks for specifying strategies and outcomes (e.g., Taxonomy of Implementation Outcomes [51], RE-AIM [95]) could help standardize language and measurement, and allow for more systematic comparisons between studies [96]. Collaboration between implementation science, QI researchers, and healthcare administrators could help address this gap [34,97].

## Limitations

There are several study limitations that should be considered. Despite a structured and systematic search, we may have missed some hospital implementation and QI projects. Implementation science is a relatively new field and so most search databases do not have implementation science-related structured vocabulary. We searched title/abstract key words such as "evidence-based intervention", "implementation", and "project", which may miss studies if implementation-specific vocabulary was not used in the title or abstract. Our search for published work may also be biased towards studies from well-resourced hospitals or academic medical centers with the ability to publish. We only included US-based studies published between 2016 and 2023. While we recognize that this may miss important strategies, we were targeting studies using US-based clinical recommendations and substance use services. We opted to broadly assess the inclusion of a racial and ethnic equity lens. In doing so, we may not have accurately assessed the extent to which studies conducted equity-oriented research. In line with scoping review methodology, we did not complete a formal assessment of bias or quality of evidence. Finally, one key limitation is the heterogeneity of the included studies. While this was an intentional aspect of our scoping methodology, it limits our ability to draw conclusions about the effectiveness of specific strategies. Future systematic or realist reviews may be better suited to evaluate the effectiveness and implementation of specific interventions in more detail.

## Conclusion

Existing studies provide state- and hospital-level administrators and policymakers with promising implementation and QI strategies that can be used to integrate substance use services into acute care settings for pregnant and birthing people. By clarifying what has been implemented, how it has been evaluated, and where key gaps remain, this review can support future implementation science and quality improvement efforts aimed at promoting equitable, evidence-informed care for pregnant and birthing people with SUD in acute care settings. Given the heterogeneity of the included studies, this review should be viewed as a foundational step to map the current landscape and identify critical gaps. Future research should focus on more rigorous evaluations of implementation strategies, measure downstream outcomes such as adoption and sustained use of substance use services, and apply a racial equity lens more explicitly.

## Supporting information

**S1 File. PRISMA-ScR checklist.**
(DOCX)

**S2 File. Data extraction tool.**
(DOCX)

**S3 File. Detailed summary of included studies.**
(DOCX)

## Author contributions

**Conceptualization:** Carla King, Gregory Laynor, Sugy Choi.

**Data curation:** Carla King, Adetayo Fawole.

**Formal analysis:** Carla King, Adetayo Fawole.

**Funding acquisition:** Carla King.

**Methodology:** Carla King, Adetayo Fawole, Gregory Laynor, Mishka Terplan, Matthew Lee, Sugy Choi.

**Software:** Gregory Laynor.

**Supervision:** Gregory Laynor, Jennifer McNeely, Mishka Terplan, Matthew Lee, Sugy Choi.

**Writing – original draft:** Carla King, Adetayo Fawole, Sugy Choi.

**Writing – review & editing:** Carla King, Adetayo Fawole, Gregory Laynor, Jennifer McNeely, Mishka Terplan, Matthew Lee, Sugy Choi.

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
