## [Decision Letter · Decision Letter 0]

9 Apr 2025

Dear Dr. King,

Thank you for submitting your manuscript to PLOS ONE. After careful consideration, we feel that it has merit but does not fully meet PLOS ONE’s publication criteria as it currently stands. Therefore, we invite you to submit a revised version of the manuscript that addresses the points raised during the review process.

**The manuscript has been evaluated by two reviewers, and their comments are available below. The reviewers have raised a number of concerns that need attention. Specifically, they requested addressing the heterogeneity of the articles used and the lack of clearly defined outcomes of interest of literature search. They also requested a discussion on the intersection of substance use disorders (SUDs) and maternal health disparities to provide additional context as well as adding more details to the methods section.**

**Could you please carefully revise the manuscript to address all comments raised?**

We look forward to receiving your revised manuscript.

Kind regards,

Zahra Al-Khateeb, Ph.D

Staff Editor

PLOS ONE

**Journal Requirements:**

1. When submitting your revision, we need you to address these additional requirements. Please ensure that your manuscript meets PLOS ONE's style requirements, including those for file naming. The PLOS ONE style templates can be found at https://journals.plos.org/plosone/s/file?id=wjVg/PLOSOne_formatting_sample_main_body.pdf and https://journals.plos.org/plosone/s/file?id=ba62/PLOSOne_formatting_sample_title_authors_affiliations.pdf 2. For studies involving third-party data, we encourage authors to share any data specific to their analyses that they can legally distribute. PLOS recognizes, however, that authors may be using third-party data they do not have the rights to share. When third-party data cannot be publicly shared, authors must provide all information necessary for interested researchers to apply to gain access to the data. (https://journals.plos.org/plosone/s/data-availability#loc-acceptable-data-access-restrictions)  For any third-party data that the authors cannot legally distribute, they should include the following information in their Data Availability Statement upon submission:a) A description of the data set and the third-party sourceb) If applicable, verification of permission to use the data setc) Confirmation of whether the authors received any special privileges in accessing the data that other researchers would not haved) All necessary contact information others would need to apply to gain access to the data 3. We note that Figure 2 in your submission contain map images which may be copyrighted. All PLOS content is published under the Creative Commons Attribution License (CC BY 4.0), which means that the manuscript, images, and Supporting Information files will be freely available online, and any third party is permitted to access, download, copy, distribute, and use these materials in any way, even commercially, with proper attribution. For these reasons, we cannot publish previously copyrighted maps or satellite images created using proprietary data, such as Google software (Google Maps, Street View, and Earth). For more information, see our copyright guidelines: http://journals.plos.org/plosone/s/licenses-and-copyright. We require you to either present written permission from the copyright holder to publish these figures specifically under the CC BY 4.0 license, or remove the figures from your submission: a. You may seek permission from the original copyright holder of Figure 2 to publish the content specifically under the CC BY 4.0 license.   We recommend that you contact the original copyright holder with the Content Permission Form (http://journals.plos.org/plosone/s/file?id=7c09/content-permission-form.pdf) and the following text:“I request permission for the open-access journal PLOS ONE to publish XXX under the Creative Commons Attribution License (CCAL) CC BY 4.0 (http://creativecommons.org/licenses/by/4.0/). Please be aware that this license allows unrestricted use and distribution, even commercially, by third parties. Please reply and provide explicit written permission to publish XXX under a CC BY license and complete the attached form.” Please upload the completed Content Permission Form or other proof of granted permissions as an "Other" file with your submission. In the figure caption of the copyrighted figure, please include the following text: “Reprinted from [ref] under a CC BY license, with permission from [name of publisher], original copyright [original copyright year].” b. If you are unable to obtain permission from the original copyright holder to publish these figures under the CC BY 4.0 license or if the copyright holder’s requirements are incompatible with the CC BY 4.0 license, please either i) remove the figure or ii) supply a replacement figure that complies with the CC BY 4.0 license. Please check copyright information on all replacement figures and update the figure caption with source information. If applicable, please specify in the figure caption text when a figure is similar but not identical to the original image and is therefore for illustrative purposes only.The following resources for replacing copyrighted map figures may be helpful: USGS National Map Viewer (public domain): http://viewer.nationalmap.gov/viewer/The Gateway to Astronaut Photography of Earth (public domain): http://eol.jsc.nasa.gov/sseop/clickmap/Maps at the CIA (public domain): https://www.cia.gov/library/publications/the-world-factbook/index.html and https://www.cia.gov/library/publications/cia-maps-publications/index.htmlNASA Earth Observatory (public domain): http://earthobservatory.nasa.gov/Landsat:
http://landsat.visibleearth.nasa.gov/USGS EROS (Earth Resources Observatory and Science (EROS) Center) (public domain): http://eros.usgs.gov/#Natural Earth (public domain): http://www.naturalearthdata.com/

Reviewers' comments:

Reviewer's Responses to Questions

**Comments to the Author**

1. Is the manuscript technically sound, and do the data support the conclusions?

Reviewer #1: Partly

Reviewer #2: Yes

2. Has the statistical analysis been performed appropriately and rigorously?

Reviewer #1: N/A

Reviewer #2: Yes

3. Have the authors made all data underlying the findings in their manuscript fully available?

Reviewer #1: Yes

Reviewer #2: Yes

4. Is the manuscript presented in an intelligible fashion and written in standard English?

Reviewer #1: Yes

Reviewer #2: Yes

**Reviewer #1:**  In this manuscript, the researchers attempted to review implementation science and quality improvement strategies to promote uptake of equitable, evidence-informed care for pregnant or birth people with SUDS in acute hospital-based settings. In addition, they wanted to identify barriers to care based on these interventions.

The overall search strategy was well designed; however, there are some methodological issues to consider.

Major areas to address include the following:

1. Studies included in this review were too heterogeneous.

The timeline for the literature search was based on publication dates of 8 different "guidelines" or "best practice recommendation documents"; however, the implementation strategies were not linked back to best practice recommendations. Meaning that the studies included in this review consisted of any intervention to address SUD among pregnant persons.

2. Specifying outcomes of interest prior to the literature search

The methods did not specify which outcomes were the focus of this review - was it changing provider knowledge, attitudes and behaviours or patient engagement in care? From the introduction, it seemed that both were desired outcomes but these are such broad categories that I would suggest narrowing down the studies to focus on one audience for interventions.

3. Table 2 contains too many columns and too much information. It does not provide enough details for the reader to understand the kind of interventions that were used in these studies. For example, type of substance use is irrelevant since only a minority focused on opioids and the rest on any substance use. So this may be a column to eliminate. Similarly, the heading of "strategy actor" was confusing.

4. Funding is from Canada but review is focused on US based interventions - this seems

Overall, this review included studies with too much heterogeneity in terms of audience for interventions and types of interventions to make meaningful summary statements. This limits the utility of this summary and not clear how it would be used to guide future implementation strategies. No definite conclusions could be made. Researchers may opt to narrow target audience to hospital providers since the focus of acuter care settings in this review was on hospital-based programs.

**Reviewer #2:** Overall Review

This is a well written and important manuscript that presents a systematic scoping review on strategies for implementing evidence-informed care for pregnant and birthing people with substance use disorders (SUDs) in acute care settings. In light of the evolving substance use crisis in the United States, this study is timely, well-structured, and contributes significantly to the literature on maternal healthcare and SUD treatment. Further, this publication can serve as a resource for clinicians and researchers to address implementation gaps in perinatal care. The paper is well written and thorough, and I have a few minor comments and suggestions for the authors. I hope the authors find them helpful.

Comments

1. Introduction. The introduction effectively outlines the problem but could benefit from more discussion on the intersection of SUDs and maternal health disparities to provide additional context.

2. Pp. 9 lines 193-194 and figure 1. The manuscript does not specify why 1,037 out of the 1,101 studies were excluded. Providing a brief explanation—such as lack of alignment with study objectives, absence of perinatal populations, or other relevant criteria—would improve clarity and consistency with the other reported exclusions.

3. Methods in general. It would be beneficial to specify how grey literature findings were integrated into the review to ensure reproducibility (e.g., How were grey literature sources evaluated for relevance and quality? Were they screened using the same inclusion/exclusion criteria as peer-reviewed studies? Were grey literature findings analyzed separately or synthesized with peer-reviewed studies?)

4. The racial equity lens assessment is commendable, but I wonder if authors could also describe how racial equity was coded (i.e. how was it identified and analyzed during data extraction? Did authors use a predefined framework (e.g., an existing racial equity assessment tool) or develop their own criteria? Was there a process to ensure consistency between reviewers in identifying racial equity considerations?).

5. Results. The manuscript notes that only 4 studies considered racial equity explicitly. It would be useful to also explore whether these studies showed implementation success.

6. The discussion effectively contextualizes findings within the broader healthcare landscape but could further explore implications for policymakers and hospital administrators.

7. Consider elaborating on sustainability challenges for SUD interventions in acute care settings. How do funding models and policy environments influence implementation success?

**Do you want your identity to be public for this peer review?** For information about this choice, including consent withdrawal, please see our Privacy Policy

Reviewer #1: No

Reviewer #2: No

---

## [Author Response · Author response to Decision Letter 1]

7 May 2025

We appreciate the thoughtful comments from the editor and the reviewers. We’ve addressed each comment below. We believe our changes will clarify the inclusion of heterogenous articles and the outcomes used in this review. We’ve also added context around the intersection of SUDs and maternal health, and details to the methods and discussion sections.

Please note, line numbers refer to the version with track changes.

Response to Reviewers

Reviewer #1: In this manuscript, the researchers attempted to review implementation science and quality improvement strategies to promote uptake of equitable, evidence-informed care for pregnant or birth people with SUDS in acute hospital-based settings. In addition, they wanted to identify barriers to care based on these interventions. The overall search strategy was well designed; however, there are some methodological issues to consider.

Major areas to address include the following:

1. Studies included in this review were too heterogeneous.

The timeline for the literature search was based on publication dates of 8 different "guidelines" or "best practice recommendation documents"; however, the implementation strategies were not linked back to best practice recommendations. Meaning that the studies included in this review consisted of any intervention to address SUD among pregnant persons.

Thank you for this comment. We agree that we’ve included a wide range of strategies and best practice recommendations. In part, this was intentional. We purposely selected scoping methods because we wanted to summarize the extent and variety of evidence for SUD treatment and care in the acute care setting, and identify gaps to aid future research and implementation. We’ve added this clarification to the introduction (lines 127-129). Throughout the introduction, methods, and results, we have also clarified the specific best practice recommendations we’re examining to help focus our study–i.e., those that provide care and treatment directly for SUDs. We exclude aspects of care that are best practices for pregnant and birthing people with SUDs, but that do not directly address SUDs (e.g., pain treatment, neonatal care/breastfeeding, contraception) (lines 157-159). We hope these edits clarify the purpose of including a variety of studies, and also bring more clarity and focus to the review.

2. Specifying outcomes of interest prior to the literature search

The methods did not specify which outcomes were the focus of this review - was it changing provider knowledge, attitudes and behaviours or patient engagement in care? From the introduction, it seemed that both were desired outcomes but these are such broad categories that I would suggest narrowing down the studies to focus on one audience for interventions.

Thank you. Because this is a scoping review, we did not specify an outcome the same way as we would if this were a systematic review. Throughout the introduction and methods sections (e.g. lines 103-108, 142-146), we have clarified that our outcome is adoption of SUD treatment and care best practices in acute care settings. To evaluate this outcome, we’ve broadly characterized strategies that have been implemented to address the research-to-practice gap in SUD treatment and care, and summarize the outcomes that were measured by the studies (i.e., changes to provider knowledge or attitudes, uptake of medications for SUD treatment). We hope this provides more focus in our review.

3. Table 2 contains too many columns and too much information. It does not provide enough details for the reader to understand the kind of interventions that were used in these studies. For example, type of substance use is irrelevant since only a minority focused on opioids and the rest on any substance use. So this may be a column to eliminate. Similarly, the heading of "strategy actor" was confusing.

We agree that Table 2 was confusing. We’ve streamlined it by removing the substance type, sample size, and strategy actor columns, and editing the descriptions in the strategies and results. We’ve included a more detailed table with sample size etc. for those who are interested in the Supporting Files (S3).

4. Funding is from Canada but review is focused on US based interventions - this seems

We appreciate this comment. The primary author is a Canadian who attends a US-based academic institute. She receives funding for her doctoral work as part of a Canadian program to support doctoral students studying at foreign institutions.

Overall, this review included studies with too much heterogeneity in terms of audience for interventions and types of interventions to make meaningful summary statements. This limits the utility of this summary and not clear how it would be used to guide future implementation strategies. No definite conclusions could be made. Researchers may opt to narrow target audience to hospital providers since the focus of acuter care settings in this review was on hospital-based programs.

Thank you again for your review. We believe your feedback has significantly strengthened our manuscript. We hope the changes outlined in point 1 help clarify the intentionality of heterogeneity but also add focus to the review. We also hope that by clarifying the best practice recommendations we were attempting to summarize, we’ve narrowed the scope and target audience to those who are considering implementation of best practices to specifically target SUD-related treatment and care for pregnant patients, or future research on the topic.

Reviewer #2: Overall Review

This is a well written and important manuscript that presents a systematic scoping review on strategies for implementing evidence-informed care for pregnant and birthing people with substance use disorders (SUDs) in acute care settings. In light of the evolving substance use crisis in the United States, this study is timely, well-structured, and contributes significantly to the literature on maternal healthcare and SUD treatment. Further, this publication can serve as a resource for clinicians and researchers to address implementation gaps in perinatal care. The paper is well written and thorough, and I have a few minor comments and suggestions for the authors. I hope the authors find them helpful.

Comments

1. Introduction. The introduction effectively outlines the problem but could benefit from more discussion on the intersection of SUDs and maternal health disparities to provide additional context.

Thank you for this feedback. We’ve added to the first paragraph (lines 52-65) and other sections in the introduction (lines 70-71, 79-82) to better reflect the intersection of SUDs with maternal health and related disparities.

2. Pp. 9 lines 193-194 and figure 1. The manuscript does not specify why 1,037 out of the 1,101 studies were excluded. Providing a brief explanation—such as lack of alignment with study objectives, absence of perinatal populations, or other relevant criteria—would improve clarity and consistency with the other reported exclusions.

Thank you for bringing this to our attention. We’ve added details to the results section (lines 237-239), noting that studies were excluded “because they did not meet the inclusion criteria (i.e., ineligible study types with no intervention or strategy, ineligible settings, ineligible interventions, and reports on clinical management of SUDs).” In JBI methodology, it is not common practice to record the reason for exclusion at the title and abstract review phase, so we don’t have specific numbers consistent with the full text exclusion section.

3. Methods in general. It would be beneficial to specify how grey literature findings were integrated into the review to ensure reproducibility (e.g., How were grey literature sources evaluated for relevance and quality? Were they screened using the same inclusion/exclusion criteria as peer-reviewed studies? Were grey literature findings analyzed separately or synthesized with peer-reviewed studies?)

Thanks for this. We’ve added additional details to the screening (lines 187-191) and extraction (line 196) sections in the methods. One reviewer initially screened conference abstracts using the same eligibility criteria as peer-reviewed studies (i.e., any conference abstracts that were US-based and tested a strategy that was implementing clinical care guidelines for SUD treatment). Conference abstracts were then imported to Covidence for review by a second reviewer (AF). Since we used scoping methodology, we didn’t assess study quality, only relevance. We used the same extraction template for published studies and grey literature.

4. The racial equity lens assessment is commendable, but I wonder if authors could also describe how racial equity was coded (i.e. how was it identified and analyzed during data extraction? Did authors use a predefined framework (e.g., an existing racial equity assessment tool) or develop their own criteria? Was there a process to ensure consistency between reviewers in identifying racial equity considerations?).

Thanks for this important comment. Because racial equity wasn’t the primary focus of this paper, we opted not to use an existing racial equity assessment tool. This is a limitation of our analysis, and we’ve added a comment in the limitations section to reflect that (lines 657-659). Instead, we structured our review of racial equity around Shelton 2021 who makes recommendations for the inclusion of structural racism in implementation science (i.e., via frameworks and models, implementing strategies that address structural racism and health equity, and community-engaged research). We’ve added more details about our extraction in the data extraction section (lines 209-217). In short, we extracted any sentences or paragraphs in the article where the authors used equity-specific language (“equity”, “structural racism”, “structural competency” anywhere in the study (explicit), or wrote about racial disparities, but did not use equity-specific language (implicit). Reviewers resolved any discrepancies via discussion.

5. Results. The manuscript notes that only 4 studies considered racial equity explicitly. It would be useful to also explore whether these studies showed implementation success.

Thank you for this comment. Since the studies were primarily descriptive, we were unable to report on implementation success related to racial equity. We’ve added a line to clarify this in the racial equity section (lines 544-545). We also included a section in the discussion about one study that provides a good example of including a racial equity lens (lines 627-631].

6. The discussion effectively contextualizes findings within the broader healthcare landscape but could further explore implications for policymakers and hospital administrators.

We agree and have added more specific implications for policymakers and hospital administrators throughout the discussion section. For example, we’ve added a section in the first paragraph that notes how hospital administrators may consider using these strategies in their healthcare systems (lines 567-572) and a note about hospital administrators collecting quality metrics in lines 634-637. We also included a section on policymakers’ role in sustainability (also in response to your comment below) in lines 604-614.

7. Consider elaborating on sustainability challenges for SUD interventions in acute care settings. How do funding models and policy environments influence implementation success?

Thank you for this. We’ve added a paragraph on sustainability that describes the importance of funding support for networks/collaboratives, improving reimbursements, and investments in dedicated staff (lines 604-614).

---

## [Decision Letter · Decision Letter 1]

25 Jun 2025

Dear Dr. King,

Thank you for submitting your manuscript to PLOS ONE. After careful consideration, we feel that it has merit but does not fully meet PLOS ONE’s publication criteria as it currently stands. Therefore, we invite you to submit a revised version of the manuscript that addresses the points raised during the review process.

**Editor's comments**

Authors are required to make all data underlying the findings described fully available, without restriction, and from the time of publication. PLOS allows rare exceptions to address legal and ethical concerns. See the PLOS Data Policy and FAQ for detailed information. Kindly follow the instructions on data availability statement and provide clear reasons as to why the data cannot be made available.

Please ensure consistency of the subject, research objective, methods, and conclusions made when revising. Your objective does mention that you are looking at strategies to "promote uptake of equitable, evidence-informed care for pregnant or birth people with SUDS in acute hospital-based settings". `From this objective we expect the focus of the paper to be on strategies to increase demand and hence uptake. Your response was that you "summarized the outcomes that were measured by the studies (i.e., changes to provider knowledge or attitudes, uptake of medications for SUD treatment)." This is not clear since your objective has specified the outcome to be "adoption/uptake". For every atrategy listed, the readers need to see how they are related to adoption/uptake of equitable, evidence-informed care for pregnant or birth people with SUDS. Even though scoping reviews cover wide topics, you are still limited to what your objectives are. Kindly revise accordingly.

We look forward to receiving your revised manuscript.

Kind regards,

Belinda J Njiro, M.D

Academic Editor

PLOS ONE

Reviewers' comments:

Reviewer's Responses to Questions

**Comments to the Author**

Reviewer #1: (No Response)

Reviewer #2: All comments have been addressed

2. Is the manuscript technically sound, and do the data support the conclusions?

Reviewer #1: Partly

Reviewer #2: Yes

3. Has the statistical analysis been performed appropriately and rigorously?

Reviewer #1: N/A

Reviewer #2: Yes

4. Have the authors made all data underlying the findings in their manuscript fully available?

Reviewer #1: Yes

Reviewer #2: Yes

5. Is the manuscript presented in an intelligible fashion and written in standard English?

Reviewer #1: Yes

Reviewer #2: Yes

Reviewer #1: Thanks for your responses and changes in response to previous comments.

1. Introduction

Thanks for the clarifications as outlined in your response letter. I appreciated the attempt to specify which recommendations were included as follows "... including SUD treatment referrals, medications for addiction treatment, harm

reduction resources and education, and non-punitive/non-stigmatizing care, could help improve

quality of care and address inequitable SUD treatment gaps. [11, 12, 30]". These are broad categories in themselves and overlapping. For example, medications to address SUD are generally considered as harm reduction strategies. I would suggest picking 2 or 3 specific recommendations such as medications or referrals to addiction treatment programs and then evaluating how these were implemented.

You stated that "We exclude aspects of care that are best practices for pregnant and birthing people with SUDs, but that do not directly address SUDs"; however, in the section on system level changes, the use of supports during labour and delivery to help with intrapartum ?pain was included which is not directly related to SUD.

2. Methodology

You have indicated that you followed the PRISM flowchart for scoping reviews; however, the checklist for this methodology still requires clear indication of "Data items in section 11 - List and define all variables for which data were sought

and any assumptions and simplifications made." Therefore, the outcomes of interest for this review would be helpful.

Overall, organizing articles into 3 categories provided a structure for this review but the 3 topics themselves are very broad. Changing provider attitudes was not linked to change in practice which would be more meaningful in terms of evidence for educational interventions or persistent change in attitudes at months post-intervention. These specifics would be more meaningful in terms of determining what has been effective and what should be changed in future interventions.

As mentioned previously, this review's broad focus leads to inclusion of studies with too much heterogeneity in terms of audience for interventions and types of interventions to make meaningful summary statements.

Reviewer #2: I thank the authors for thoroughly addressing reviewer comments and editing the manuscript thoughtfully.

**Do you want your identity to be public for this peer review?** For information about this choice, including consent withdrawal, please see our Privacy Policy

Reviewer #1: No

Reviewer #2: No

---

## [Author Response · Author response to Decision Letter 2]

18 Jul 2025

Dr. Belinda J Njiro, M.D.

Academic Editor

PLOS ONE

Dear Dr. Njiro,

Thank you for the thorough and thoughtful feedback from you and the reviewers. Your comments have helped us clarify our subject, objectives, and be consistent throughout the manuscript. After carefully reflecting on the comments, we re-organized our results to more closely align with the objectives of the paper. The results are now organized into 5 types of strategies that were implemented in acute care settings for substance use disorder care for pregnant and birthing people: 1) education and learning collaboratives; 2) clinical pathways or workflows; 3) brief interventions; 4) peer support; and 5) structural changes. In response to reviewer #1, we’ve explicitly stated our outcomes of interest and re-organized our results in hopes of addressing issues with study heterogeneity and utility of the review. We hope this brings clarity and meaning to our conclusions.

We’ve addressed each comment more specifically in our response to reviewers. All page and line numbers refer to the version with tracked changes hidden. Thank you for considering our revised manuscript.

Sincerely,

Carla King, MPH, PhD candidate

RESPONSE TO REVIEWERS

Editor's comments

1. Data availability statement

Authors are required to make all data underlying the findings described fully available, without restriction, and from the time of publication. PLOS allows rare exceptions to address legal and ethical concerns. See the PLOS Data Policy and FAQ for detailed information. Kindly follow the instructions on data availability statement and provide clear reasons as to why the data cannot be made available.

We’ve added a data availability statement on page 37 (line 955) stating “All data analyzed in this review are from previously published studies, which are cited throughout the manuscript. No new data were generated.”

2. Please ensure consistency of the subject, research objective, methods, and conclusions made when revising. Your objective does mention that you are looking at strategies to "promote uptake of equitable, evidence-informed care for pregnant or birth people with SUDS in acute hospital-based settings". From this objective we expect the focus of the paper to be on strategies to increase demand and hence uptake. Your response was that you "summarized the outcomes that were measured by the studies (i.e., changes to provider knowledge or attitudes, uptake of medications for SUD treatment)." This is not clear since your objective has specified the outcome to be "adoption/uptake". For every strategy listed, the readers need to see how they are related to adoption/uptake of equitable, evidence-informed care for pregnant or birth people with SUDS. Even though scoping reviews cover wide topics, you are still limited to what your objectives are. Kindly revise accordingly.

Thank you for this comment. It was very helpful in guiding how we clarified and aligned our objective, outcomes, and overall framing. In response, we have edited relevant sections throughout the manuscript to ensure consistency across subject, objective, methods, results and conclusions.

We clarified that the subject of our review is the integration of SUD services (i.e., medications, referrals to treatment, and overdose prevention) into acute care settings for pregnant and birthing people (page 4). We also updated the manuscript’s title to more clearly articulate this: “Integrating substance use disorder services into acute care settings for pregnant and birthing people: A systematic scoping review of implementation and quality improvement strategies”

Previously, our use of the term “uptake/demand” in the objective was ambiguous and did not fully reflect the focus of our analysis. We’ve more clearly stated our objective on page 5:

“To address this gap, we used scoping methods to systematically review studies that used an implementation or QI strategy to integrate SUD services for pregnant or birthing people into an acute care setting. Scoping methods were chosen to review a body of research that is in its early stage of development, and to identify gaps in measurement and outcomes that would aid future research and implementation. [55] We aimed to 1) characterize and describe the implemented strategies; 2) assess the inclusion of racial equity in study design and implementation; 3) summarize measures and outcomes used to evaluate implementation; and 4) identify reported barriers and facilitators to implementation, sustainability, and/or scale-up.”

To align with this objective, we’ve also more explicitly specified our outcomes on page 8:

“Our outcomes of interest were 1) implementation-related; and 2) the extent to which a racial and ethnic equity lens was incorporated into the study design and implementation. We extracted any reported implementation outcomes that related to SUD services for pregnant and birthing people, including acceptability, feasibility, adoption, and sustainability (see S2 File). [65] We also noted the level at which the outcome was measured, when applicable (i.e., patient-, provider-, hospital-, or community-level). To assess how studies acknowledged and addressed racial and ethnic equity in their design and implementation, we extracted explicit or implicit mentions of racial equity using Shelton et al.’s recommendations for addressing racism in implementation science as a guide…”

We’ve also revised Table 2 to include columns with both outcomes of interest (i.e., implementation-related outcomes and racial equity in design/implementation).

We thank you again for this comment and hope these changes more clearly align the manuscript with our stated objective and research focus.

Reviewer #1

Thanks for your responses and changes in response to previous comments.

1. Introduction

Thanks for the clarifications as outlined in your response letter. I appreciated the attempt to specify which recommendations were included as follows "... including SUD treatment referrals, medications for addiction treatment, harm reduction resources and education, and non-punitive/non-stigmatizing care, could help improve quality of care and address inequitable SUD treatment gaps. [11, 12, 30]". These are broad categories in themselves and overlapping. For example, medications to address SUD are generally considered as harm reduction strategies. I would suggest picking 2 or 3 specific recommendations such as medications or referrals to addiction treatment programs and then evaluating how these were implemented.

Thank you again for your review and helpful comments. Based on your suggestion, we’ve narrowed our focus to specific clinical practices: medications, referrals to treatment, and overdose prevention. Through the manuscript, we refer to these as “SUD services”. While we recognize this still covers a broad range of SUD practices, we feel that these are often overlapping and implemented together. We hope that by limiting our review to these 3 key elements, we have more clearly evaluated how they were implemented.

You stated that "We exclude aspects of care that are best practices for pregnant and birthing people with SUDs, but that do not directly address SUDs"; however, in the section on system level changes, the use of supports during labour and delivery to help with intrapartum ?pain was included which is not directly related to SUD.

Thanks for this comment. Perhaps we created confusion because we didn’t define “plans of safe care” or provide enough details about the hospital policy by Martin et al. described in this section. In response, we’ve added a line describing the purpose of plans of safe care:

“Another study evaluated the state’s implementation of the Comprehensive Addiction Recovery Act (CARA) and Plans of Safe Care which are intended to provide supportive, non-punitive SUD treatment referrals and care coordination for pregnant people with SUDs and their families” Page 21, Line 406-409.

We’ve also added more information about the hospital policy study we included:

“The revised hospital policy recommended hospital providers respond to patients, including those who are pregnant and racialized, who were using substances during hospitalization by offering pain and withdrawal support, addiction treatments, and supportive services for the SUD.” Page 21, Line 398-401.

We hope these changes address your comment.

2. Methodology

You have indicated that you followed the PRISM flowchart for scoping reviews; however, the checklist for this methodology still requires clear indication of "Data items in section 11 - List and define all variables for which data were sought and any assumptions and simplifications made." Therefore, the outcomes of interest for this review would be helpful.

Thank you for bringing this to our attention.

We’ve added our outcomes of interest to the data extraction section on page 8 (these are also listed in Supplementary File 2):

“Our outcomes of interest were 1) implementation-related; and 2) the extent to which a racial and ethnic equity lens was incorporated into the study design and implementation. We extracted any reported implementation outcomes that related to SUD services for pregnant and birthing people, including acceptability, feasibility, adoption, and sustainability (see S2 File). [65] We also noted the level at which the outcome was measured, when applicable (i.e., patient-, provider-, hospital-, or community-level). To assess how studies acknowledged and addressed racial and ethnic equity in their design and implementation, we extracted explicit or implicit mentions of racial equity using Shelton et al.’s recommendations for addressing racism in implementation science as a guide."

We also updated the PRISMA-ScR checklist to reflect the correct page numbers.

Overall, organizing articles into 3 categories provided a structure for this review but the 3 topics themselves are very broad. Changing provider attitudes was not linked to change in practice which would be more meaningful in terms of evidence for educational interventions or persistent change in attitudes at months post-intervention. These specifics would be more meaningful in terms of determining what has been effective and what should be changed in future interventions.

As mentioned previously, this review's broad focus leads to inclusion of studies with too much heterogeneity in terms of audience for interventions and types of interventions to make meaningful summary statements.

Thank you again for this comment – it has significantly helped improve the clarity and organization of our manuscript. After carefully reflecting on your comments, we reorganized our results to better align with the objectives of the paper (i.e., characterizing strategies used to integrate SUD services). The results are now organized into 5 groups reflecting the type of strategy that was implemented: 1) education and learning collaboratives; 2) clinical pathways or workflows; 3) brief interventions; 4) peer support; and 5) structural changes. To guide this reclassification, we employed the Expert Recommendations for Implementing Change framework (referenced on page 9).

We hope this reorganization will help readers better understand the types of strategies used, the outcomes that were and were not measured, and where next steps are needed to assess effectiveness, adoption etc. We recognize that these strategies themselves can be applied to multiple aspects of SUD service delivery and intentionally include a range to illustrate what has been implemented and what gaps remain. Given this is a scoping review, we hope this broadly covers the extent of variability so that more targeted, bundled interventions can be studied or implemented.

We also revised the discussion to clarify the intended use of this review as a foundational step toward more targeted implementation science and quality improvement efforts, and added to our limitations section on page 30, lines 631-634 to more directly address study heterogeneity:

“Finally, one key limitation is the heterogeneity of the included studies. While this was an intentional aspect of our scoping methodology, it limits our ability to draw conclusions about the effectiveness of specific strategies. Future systematic or realist reviews may be better suited to evaluate the effectiveness and implementation of specific interventions in more detail.”

We hope our changes help clarify the utility and limitations of this review.

Reviewer #2: I thank the authors for thoroughly addressing reviewer comments and editing the manuscript thoughtfully.

We thank you again for your review.

---

## [Decision Letter · Decision Letter 2]

7 Jan 2026

Dear Dr. King,

Thank you for submitting your manuscript to PLOS ONE. After careful consideration, we feel that it has merit but does not fully meet PLOS ONE’s publication criteria as it currently stands. Therefore, we invite you to submit a revised version of the manuscript that addresses the points raised during the review process.

We look forward to receiving your revised manuscript.

Kind regards,

Abid Rizvi

Academic Editor

PLOS One

Journal Requirements:

Reviewers' comments:

Reviewer's Responses to Questions

**Comments to the Author**

Reviewer #1: All comments have been addressed

Reviewer #2: All comments have been addressed

2. Is the manuscript technically sound, and do the data support the conclusions?

Reviewer #1: Yes

Reviewer #2: Yes

3. Has the statistical analysis been performed appropriately and rigorously?

Reviewer #1: N/A

Reviewer #2: Yes

4. Have the authors made all data underlying the findings in their manuscript fully available?

Reviewer #1: Yes

Reviewer #2: Yes

5. Is the manuscript presented in an intelligible fashion and written in standard English?

Reviewer #1: Yes

Reviewer #2: Yes

Reviewer #1: Thank you to the authors for addressing the reviewer comments and making the indicated revisions to this manuscript.

Overall, the clarifying points made this manuscript easier to follow. However, there are a couple of brief feedback points that I would like to highlight.

1. Title: thanks for revising the title; however, I am not sure if "Integrating" is the correct descriptive word of the strategies, the description oft he types of services lead to perhaps "implementing" or "adopting" since some of the strategies occurred only once (eg. educational initiatives).

2. Introduction: The intro section is very long - 3 page and there are many facts presented; however, it would be easier to follow where the article is going by having a more cohesive presentation.

The terminology "acute care settings" sounds confusing since the main focus is hospital inpatient care so why not label it as such. Acute care settings refer to both outpatient and inpatient settings, but this review is aimed at inpatient hospital setting primarily.

3. Outcomes under data extraction are mentioned very vaguely. It appears that there were no predefined outcomes and that researchers accepted all published outcomes of various strategies.

Grouping studies into five types makes the vast amount of information provided much easier to follow and to understand. Overall, there is a heterogeneous mix of services and strategies presented which suggests a wide variation in types of strategies even within one category which lends to difficulty in future studies.

Reviewer #2: (No Response)

**Do you want your identity to be public for this peer review?** For information about this choice, including consent withdrawal, please see our Privacy Policy

Reviewer #1: No

Reviewer #2: No

---

## [Author Response · Author response to Decision Letter 3]

8 Jan 2026

RESPONSE TO REVIEWERS

Reviewer #1

Thank you to the authors for addressing the reviewer comments and making the indicated revisions to this manuscript.

Overall, the clarifying points made this manuscript easier to follow. However, there are a couple of brief feedback points that I would like to highlight.

1. Title: thanks for revising the title; however, I am not sure if "Integrating" is the correct descriptive word of the strategies, the description of the types of services lead to perhaps "implementing" or "adopting" since some of the strategies occurred only once (eg. educational initiatives).

Thank you again for your review and suggestions. We have replaced “integrating” with “implementing” in the title and throughout the manuscript.

2. Introduction: The intro section is very long - 3 page and there are many facts presented; however, it would be easier to follow where the article is going by having a more cohesive presentation.

The terminology "acute care settings" sounds confusing since the main focus is hospital inpatient care so why not label it as such. Acute care settings refer to both outpatient and inpatient settings, but this review is aimed at inpatient hospital setting primarily.

Thank you for this feedback. We have made edits to the content and structure of the introduction, while maintaining a previous reviewer’s suggestion to add more depth. We have also defined “acute care settings” earlier in the introduction, specifying that acute care in our study refers to inpatient, labor/delivery, or emergency care only. Although there were only a few studies that focused on the emergency department, we felt that specifying hospital inpatient care did not accurately describe those studies, and so opted to keep “acute care”. We hope that by defining it earlier in the introduction, we have made this less confusing.

3. Outcomes under data extraction are mentioned very vaguely. It appears that there were no predefined outcomes and that researchers accepted all published outcomes of various strategies.

Thanks for this comment. You are correct that we did not pre-define specific outcomes. Because we used scoping methodology, we examined all implementation outcomes, as defined by Proctor et al. 2011, in the included studies. We have added clarity in this section (lines 179-183) by referencing the framework we used to define implementation outcomes:

“We extracted reported implementation outcomes, as defined by Proctor et al., that related to implementation of substance use services for pregnant and birthing people, including acceptability, feasibility, adoption, and sustainability (see S2 File). [52] We also noted the level at which the outcome was measured, when applicable (i.e., patient-, provider-, hospital-, or community-level).

4. Grouping studies into five types makes the vast amount of information provided much easier to follow and to understand. Overall, there is a heterogeneous mix of services and strategies presented which suggests a wide variation in types of strategies even within one category which lends to difficulty in future studies.

Thanks for this comment. We agree – there is a heterogeneous mix of services and strategies presented. While this is noted as a limitation of our study (lines 630-634), we also believe this is an important finding.

To this end, we have made recommendations for future studies throughout our discussion and conclusion sections. For example, in our discussion, we emphasized that there are many heterogenous interventions that focus on shifting provider attitudes and knowledge, however, few have successfully evaluated which of these strategies led to sustained adoption or improved patient outcomes, and so more rigorous studies are needed to determine how to best use these strategies (lines 529-534; 551-558; 570-574). We also noted that there was wide variability in how implementation outcomes were defined and measured within and across strategies limiting comparisons. In response, we have provided recommendations for future research on lines 607-614:

Strategies and implementation outcomes were often poorly defined and heterogeneous, making it difficult to compare across studies or to determine mechanisms that made a strategy successful within a specific context. Using existing implementation frameworks for specifying strategies and outcomes (e.g., Taxonomy of Implementation Outcomes [51], RE-AIM [95]) could help standardize language and measurement, and allow for more systematic comparisons between studies. [96] Collaboration between implementation science, QI researchers, and healthcare administrators could help address this gap. [34, 97]

Overall, we believe our study identifies a clear gap in the current literature and recommend that future research focuses on more clearly defined interventions and outcomes so that categories of strategies can be more rigorously evaluated and compared (lines 643-647). We hope this has addressed your concerns regarding the heterogeneity of strategies. Thank you again for your thoughtful and thorough feedback that has improved our manuscript.

Reviewer #2

(No Response)

---

## [Editor Report · Decision Letter 3]

19 Feb 2026

Implementing substance use services into acute care settings for pregnant and birthing people: A systematic scoping review of implementation and quality improvement strategies

PONE-D-24-60102R3

Dear Dr. King,

We’re pleased to inform you that your manuscript has been judged scientifically suitable for publication and will be formally accepted for publication once it meets all outstanding technical requirements.

Kind regards,

Kingston Rajiah

Academic Editor

PLOS One

Additional Editor Comments (optional):

The authors have addressed the reviewers comments
---

## [Editor Report · Acceptance letter]

PONE-D-24-60102R3

PLOS One

Dear Dr. King,

I'm pleased to inform you that your manuscript has been deemed suitable for publication in PLOS One. Congratulations! Your manuscript is now being handed over to our production team.

Kind regards,

on behalf of

Dr Kingston Rajiah

Academic Editor

PLOS One